# Shape-Aware Adversarial Learning for Scribble-Supervised Medical Image Segmentation with a MaskMix Siamese Network: A Case Study of Cardiac MRI Segmentation

**DOI:** 10.3390/bioengineering11111146

**Published:** 2024-11-13

**Authors:** Chen Li, Zhong Zheng, Di Wu

**Affiliations:** College of Computer Science and Technology, National University of Defense Technology, Changsha 410073, China; lichen14@nudt.edu.cn (C.L.); wudi09@nudt.edu.cn (D.W.)

**Keywords:** medical image segmentation, shape-aware adversarial learning, scribble annotation, siamese network, cardiac MRI

## Abstract

The transition in medical image segmentation from fine-grained to coarse-grained annotation methods, notably scribble annotation, offers a practical and efficient preparation for deep learning applications. However, these methods often compromise segmentation precision and result in irregular contours. This study targets the enhancement of scribble-supervised segmentation to match the accuracy of fine-grained annotation. Capitalizing on the consistency of target shapes across unpaired datasets, this study introduces a shape-aware scribble-supervised learning framework (MaskMixAdv) addressing two critical tasks: (1) Pseudo label generation, where a mixup-based masking strategy enables image-level and feature-level data augmentation to enrich coarse-grained scribbles annotations. A dual-branch siamese network is proposed to generate fine-grained pseudo labels. (2) Pseudo label optimization, where a CNN-based discriminator is proposed to refine pseudo label contours by distinguishing them from external unpaired masks during model fine-tuning. MaskMixAdv works under constrained annotation conditions as a label-efficient learning approach for medical image segmentation. A case study on public cardiac MRI datasets demonstrated that the proposed MaskMixAdv outperformed the state-of-the-art methods and narrowed the performance gap between scribble-supervised and mask-supervised segmentation. This innovation cuts annotation time by at least 95%, with only a minor impact on Dice performance, specifically a 2.6% reduction. The experimental outcomes indicate that employing efficient and cost-effective scribble annotation can achieve high segmentation accuracy, significantly reducing the typical requirement for fine-grained annotations.

## 1. Introduction

Medical image segmentation usually involves the painstaking labelling of medical images, which can be both time-consuming and labor-intensive. Additionally, it often requires the expertise of experienced doctors, which is sometimes in short supply. To address the challenges posed by the high cost of labelling and technical complexities, medical image segmentation has transitioned from the paradigm of fine-grained annotation to the use of coarse-grained alternatives [1,2,3]. Compared to traditional fine-grained mask annotations, scribbles are simpler and less precise inputs, typically drawn over the segmentation targets. They provide a rough indication of the object’s location without the need for detailed boundary annotations. Therefore, scribble annotation stands out as one of the most practical, efficient, and flexible coarse-grained methods, which finishes annotation in just a few seconds per image [4]. And it has become a preferred method for preparing datasets for medical image segmentation, especially for scenarios where expert time is limited and the need for large, well-annotated datasets is high.

However, constrained by the uncertainty of vast unlabelled pixels, scribble annotations lack critical positional information about the object structure, thus raising the difficulty of training neural network compared to mask annotations. Therefore, the performance of scribble-supervised medical image segmentation falls far short of those achieved by mask-supervised segmentation counterpart due to two main challenges: (1) accurately locating regions of interest (ROI) boundaries, and  (2) refining irregular shapes in segmentation predictions. When these challenges are not addressed, the generated pseudo labels can often conflict with the actual contours of ROI. This can lead to pseudo label-based model fine-tuning being misleading and uncertain.

This study aims to address these challenges and enhance the performance of scribble-supervised segmentation. It has been observed that shape of the specific segmentation target is often shared among different datasets of the same type. Incorporating shared shape priors from existing datasets may help refine the boundaries of segmentation predictions in the dataset being analyzed. In light of the aforementioned considerations, this study introduces a shape-aware scribble-supervised learning framework called MaskMixAdv for medical image segmentation. This framework centers around two core tasks that mutually enhance each other, following a similar concept to generative adversarial learning [5]: (1) Pseudo label generation: A mixup-based masking strategy (MaskMix) introduces image-level or feature-level data augmentation to enrich scribbles. In this process, a dual-branch siamese network is designed as the backbone to generate fine-grained pseudo labels through scribble-supervised learning; (2) Pseudo label optimization: A shared shape prior is extracted from external datasets containing the same ROI but with unpaired mask annotations through adversarial learning. In this process, a CNN-based discriminator distinguishes the generated pseudo labels from real masks, together regularizing the contours of pseudo labels towards model fine-tuning.

Experiments evaluated the performance of the proposed MaskMixAdv framework in cardiac MRI segmentation. Experimental results of MaskMixAdv outperformed the state-of-the-art methods in scribble-supervised cardiac MRI segmentation, achieving 88.7% and 88.6% average Dice score in two datasets.

The main contributions of this work are summarized as follows: (1) This study has successfully explored an annotation-efficient approach to dataset preparation for medical image segmentation, taking cardiac MRI segmentation as a case study. It achieves comparable performance to pixel-by-pixel labeling methods, but without the time-consuming process. (2) The study introduces a shape-aware adversarial learning framework that generates segmentation predictions accurately representing real shapes. This framework effectively refines pseudo labels by transferring shape priors from additional datasets with matching regions of interest (ROI). (3) This study introduces a novel masking approach that enables single-sample data augmentation, addressing the need to enrich coarse-grained scribble annotations. This method offers a viable alternative to existing Mixup-based cross-sample data augmentation methods.

The remainder of this paper is organized as follows: Section 2 introduces the most salient works along scribble-supervised medical image segmentation. Section 3 details the proposed shape-aware scribble-supervised learning framework. Section 4 describes the experiments to evaluate the capabilities of the proposed framework and provides the results and analyses. Section 5 concludes this study and provides a plan for future work.

## 2. Related Work

Scribble-supervised learning has garnered increasing attention in the machine learning community due to its annotation-efficient approach to medical image segmentation, leveraging coarse-grained scribble annotations. Research in this field primarily focuses on generating dense, pixel-wise predictions that closely resemble masks derived from these annotations. The most salient works along scribble-supervised medical image segmentation can be categorized as either graphical-based or neural network-based, as detailed below.

Graphical -based scribble-supervised methods typically involve two iterative steps: (1) augmenting coarse-grained scribbles to relabel unknown pixels, and (2) training a segmentation model with the relabelled datasets. In this direction, graphical-based machine learning methods are widely used for pre-processing scribbles. For instance, Lin et al. [6] proposed a GrabCut [7]-based method called ScribbleSup to enlarge annotations from scribbles to unlabelled pixels. Vernaza et al. [8] integrated Random Walker [9] with edge detection and estimated the uncertainty when propagating labelled pixels to unknown regions. Tang et al. [10] integrated the Conditional Random Fields (CRF) [11]-based refining into the model training with a KernelCut [12]-based regularization loss. These early methods were primarily developed for natural image segmentation, with few applications in medical image segmentation.

With the growth of advanced neural networks, the exploration of graphical-based pre-processing methods in conjunction with neural networks and their application to medical image segmentation tasks were emerging. Can et al. [13] explored scribble-supervised prostate MRI segmentation, consisting of Random Walker [9] as a pre-processing step for generating seed area via region growing, and dense CRF [14] as a post-processing step for joint recurrent neural network training. They achieved 0.772 Dice on prostate segmentation. Pu et al. [15] utilized a graph convolutional network (GraphNet) to construct a graph structure for each input image at the beginning. GraphNet then augmented the unlabelled pixels by extracting semantic content near the labelled scribble with high-confidence in the graph as new labels. Ji et al. [16] explored scribble-supervised brain tumor MRI segmentation by integrating GraphCut [17] and U-Net [18]. GraphCut augmented the labelled pixel in scribbles and U-Net was trained with partial cross-entropy loss and dense CRF loss. They achieved 0.654 Dice on enhanced tumor segmentation. These methods significantly advanced scribble-supervised image segmentation by employing a two-step approach derived from graphical-based methods.

Recent neural network-based studies focused on implementing scribble-supervised medical image segmentation in an end-to-end manner. These methods often start by enlarging coarse-grained scribble annotations using Mixup [19]-based augmentation techniques such as CycleMix [20], Co-mixup [21], Cutout [22], CutMix [23], and Puzzle Mix [24]. Neural networks then generate “pseudo labels” directly during training, which are used for model fine-tuning [25,26,27], eliminating the need for relabelling datasets via graphical-based methods. Lee et al. [28] explored the adaptation of scribble-supervised learning in cell histopathology image segmentation, achieving comparable performance to mask-supervised counterpart. Built upon the U-Net [18] with ResNet-50 [29] encoder, this study utilized the exponential moving average strategy to update pseudo labels at each epoch from scribble annotations. Liu et al. [30] proposed a teacher-student model (USTM-Net) to segment COVID-19 infection from CT images. USTM-Net was trained with transformation consistency and partial cross-entropy loss, achieving 0.723 Dice on infection lesions. It demonstrated that the effectiveness of transformation consistency in enlarging sparse supervisions. Luo et al. [4] applied scribble-supervised learning into cardiac MRI segmentation, achieving 0.872 Dice. They proposed an auxiliary segmentation branch network (WSL4MIS) to generate pseudo labels by integrating the segmentation predictions from two branches randomly. It demonstrated that the advantages of multi-branch neural network in generating pseudo labels.

In addition to scribble-supervised segmentation techniques, there existed a broader category of weakly-supervised segmentation methods in general. These encompassed approaches designed for bounding boxes and image-level labels. For instance, Kulharia et al. [31] had developed a method that calculated the filling rate of each class within annotated bounding boxes to inform the training of a segmentation model. They employed CAM-based maps to diminish attention to background regions and concentrate on foreground objects within the bounding boxes. Their method achieved a mean Intersection over Union (mIoU) of 76.4% on the PASCAL VOC 2012 validation set, surpassing previous weakly-supervised techniques. Cholakkal et al. [32] introduced an image-level-supervised approach to generate a density map for object counting and instance segmentation. This was achieved under the framework of an image classification branch and a density estimation branch. With only image-level supervision, their method enhanced the state-of-the-art performance from 37.6 to 44.3 in terms of average best overlap, realizing a relative improvement of 17.8% on the PASCAL VOC 2012 dataset.

Besides, there were some studies focus on the shape regularization for weak supervision. Zhang et al. [33] explored to constrain shape of pseudo labels in scribble-supervised cardiac MRI segmentation, obtaining 0.851 Dice. They utilized the EM algorithm [34] to constrain the proportion of each class in segmentation predictions to match its in the unlabelled pixels. It demonstrated that regularizing the shape of pseudo labels contributed to improving segmentation performance. Adiga et al. [35] introduced a novel approach for estimating segmentation uncertainty by learning an anatomy-aware representation to emulate available segmentation masks. This representation aided in mapping new segmentation predictions to anatomically plausible partitions. However, the used anatomy-aware models required precise annotated data to learn the consistency of anatomical structures, making them unsuitable for directly applying to sparsely annotated datasets. Furthermore, while these models performed well on the same source dataset, their generalization capabilities may be limited when encountering different datasets or anatomical structures.

Inspired by these works, this study aims at scribble-supervised medical image segmentation with a focus on: (1) enlarging coarse-grained annotations, (2) enhancing scribble-supervised learning, and (3) optimizing irregular pseudo labels.

## 3. MaskMixAdv: Framework Enabling Scribble-Supervised Medical Image Segmentation

This section first presents the overall design of the MaskMixAdv framework. It then details the two core tasks of MaskMixAdv: (1) pseudo label generation with MaskMix masking upon a dual-branch siamese network, and (2) pseudo label optimization with the shape-aware adversarial learning.

### 3.1. Overall Design of the MaskMixAdv Framework

Figure 1 intuitively compares mask-supervised segmentation, scribbles-supervised segmentation, and the proposed adversarial scribbles-supervised segmentation. Figure 1c shows the high-level architecture of the MaskMixAdv framework. The MaskMixAdv framework works within an adversarial scribble-supervised segmentation paradigm, enabling access to (1) the randomly sampled batch (*B*) from source dataset, i.e., medical images $X\in R^{B\times H\times W\times C}$ and their paired scribble annotations $Y\in R^{B\times H\times W}$, and  (2) another randomly sampled batch (*B*) from external datasets, i.e., masks $Z\in R^{B\times H\times W}$ with the same ROI but unpaired to the *X*. The objective is to perform pixel-by-pixel dense prediction $\hat{Y}\in R^{B\times H\times W}$ of the sampled medical images *X* and give high-quality boundaries of the region of interest.

Figure 2 gives the detailed architecture of the proposed MaskMixAdv framework, where a dual-branch siamese network (DBSN) serves as backbone (θe, $\theta d_{0}$, $\theta d_{1}$, $\theta d_{2}$) for scribble-supervised learning (detailed in Section 3.2.1). Based on the dual-branch siamese network, a novel Mixup-based Masking strategy (MaskMix) is proposed to performs data augmentation (detailed in Section 3.2.2). An additional CNN-based discriminator ($\theta_{dis}$) is connected to the siamese network for shape-aware adversarial learning (detailed in Section 3.3). Based on the proposed backbone and discriminator, MaskMixAdv works in two phases:MaskMix phase receives medical image *X* and paired scribble annotations *Y* as input. In this phase, the proposed MaskMix strategy first carries out masking augmentation towards *X* at the image-level or feature-level. Then, the shared encoder θe extracts perturbed features from *X* after augmentation. Next, two independent decoders $\theta d_{1}$, $\theta d_{2}$ perform scribble-supervised training with *Y* and obtain coarsely segmentation results $Y_{seg1},Y_{seg2}$. Meanwhile, another decoder ($\theta d_{0}$) is used for masked image modeling from perturbations. Finally, MaskMix integrates $Y_{seg1}$ and $Y_{seg2}$ to generate pseudo labels $Y_{pse}$, which is sent to the next phase for model fine-tuning.Adversarial phase receives pseudo labels $Y_{pse}$ and unpaired mask *Z* as input. In this phase, *Z* is regarded as *real* and $Y_{pse}$ is labelled as *fake*. Then, the discriminator $\theta_{dis}$ receives $Y_{pse}$ or *Z* as input and outputs a bool prediction $Y_{dis}$ representing the prediction of the input source, i.e., whether it is *real* or *fake*. An adversarial loss is designed to minimize the above discrepancy of binary predictions, optimizing the training of segmentation network and discriminator. Finally, the refined pseudo labels $Y_{pse}$ participant in model fine-tuning as ground truth to optimize segmentation network.

It is worthynoting that these two phases are integrated as a whole and boost each other in a zero-sum game, just like generative adversarial learning [5]. The goal is to make it challenging for $\theta_{dis}$ to distinguish between the generated pseudo labels $Y_{pse}$ and the real masks *Z*. This way, the target shape prior can be transferred from an external dataset beforehand to scribble-supervised segmentation.

### 3.2. Pseudo Label Generation: MaskMix Masking upon Siamese Network

Previous methods struggle to extract dense target semantics directly from sparse annotation for dense prediction training. As a remedy, this study designs a dual-branch siamese network for generating pseudo labels from scribbles and proposes a data augmentation strategy (MaskMix).

The siamese network receives an input <*X*,*Y*> and then carries out scribble-supervised learning in the following five steps, whose objective is to coarsely train the siamese network, expecting to extract high-quality representations from sparse annotations to generate dense pseudo labels $Y_{pse}\in R^{B\times H\times W}$.

Data augmentation: A novel masking strategy (MaskMix) is proposed to add perturbations into the siamese network and generate two complementary binary masks, which are used for masking the input *X* at image-level or feature-level (See Section 3.2.2 for more details).Features extraction: The shared encoder ($\theta_{e}$) extracts the perturbed features of *X*, which are then used for masked reconstruction via the $\theta d_{0}$ and dense prediction via the $\theta d_{1}$ and $\theta d_{2}$ (See Section 3.2.1 for more details).Coarsely training: The extracted the features from shared encoder ($\theta_{e}$) are then sent to two segmentation decoders ($\theta d_{1}$ and $\theta d_{2}$) for scribble-supervised coarsely training. The outputs are the pixel-wise segmentation predictions $Y_{seg1},Y_{seg2}\in R^{B\times H\times W}$ with *N* classes (See Section 3.2.3 for more details).Masked image modeling: The proposed siamese network also supports self-supervised masked image modeling by reconstructing the input images *X* from MaskMix-based masking via reconstruction branch ($\theta_{e},\theta d_{0}$). The output is the reconstruction prediction $Y_{rec}\in R^{B\times H\times W\times C}$ with the same resolution as *X*. This step aims to optimize the the proposed siamese network for feature extraction from sparse supervision (See Section 3.2.4 for more details).

#### 3.2.1. Dual-Branch Siamese Network for Scribble-Supervised Learning

This section describes the proposed dual-branch siamese network (DBSN), which consists of one shared encoder $\theta_{e}$ for feature extraction and three siamese decoders ($\theta d_{0}$, $\theta d_{1}$, $\theta d_{2}$). In order to use coarse-grained labelled pixels in the scribble fully, based on the above encoder and decoders, there are a reconstruction branch and a segmentation branch inside the proposed dual-branch siamese network. Figure 3 shows the architecture:Segmentation branch consists of the shared encoder $\theta_{e}$ and two independent decoders $\theta d_{1}$, $\theta d_{2}$. This branch outputs mask-like predictions $Y_{seg1}$, $Y_{seg2}$, respectively. Above two predictions take part in the following scribble-supervised coarsely training (Section 3.2.3) and pseudo label-supervised finely training (Section 3.3.2).Reconstruction branch consists of the shared encoder $\theta_{e}$ and one independent decoder $\theta d_{0}$. This branch restores the original image from perturbations of $M_{1},M_{2}$ and then outputs image-like reconstructions $Y_{rec1},Y_{rec2}$, respectively. Above two predictions take part in the self-supervised masked image modeling (See Section 3.2.4 for more details).

Specifically, the DBSN network adopts the vanilla 2D U-Net [18] as the encoder ($\theta_{e}$)–decoder ($\theta d_{1}$) architecture, where two auxiliary decoders ($\theta d_{0},\theta d_{2}$) are extended for the reconstruction branch and segmentation branch, respectively. Additionally, a reconstruction head ($H_{rec}$) and two segmentation head ($H_{seg1}$ and $H_{seg2}$) are attached to the decoders $\theta d_{0}$, $\theta d_{1}$ and $\theta d_{2}$, respectively. Above three heads share the same architecture with different output channels.

#### 3.2.2. MaskMix: A Mixup-Based Masking and Pseudo Label Generating Strategy

The Mixup-based Masking strategy (MaskMix) is proposed to augment the sparse annotations, which consists of two steps: (1) the Mask step to add perturbations at image-level or feature-level, and (2) the Mix step to generate pseudo labels by ensembling the two segmentation branches based on the Mask step.

The Mask step introduces additional image-level or feature-level perturbations by masking the input images or extracted features, respectively. Specifically, two binary masks ($M_{1},M_{2}$) are randomly generated, where the size of $M_{1}$ and $M_{2}$ is adaptive to the size of input image or encoded feature, respectively. Besides, the two masks are complementary, i.e., $M_{1}+M_{2}$ equals a matrix consisting of all 1. Then ($M_{1},M_{2}$) are used to add perturbations:At the image level: masks ($M_{1},M_{2}$) are multiplied with images *X*. The regions of the mask ($M_{1}$ or $M_{2}$) corresponding to 1 keep the pixel values of *X* unchanged, while the other regions are filled with 0. Then these masked images ($M_{1}\cdot X,M_{2}\cdot X$) are fed into encoder $\theta_{e}$ for pixel-wise perturbation. Figure 4a illustrates the workflow of Mask step at the image-level. Figure 4c–k compare other common data perturbation strategies at the image-level.At the feature level: images *X* are directly fed into encoder $\theta_{e}$ to obtain the encoded feature $\theta_{e}\left(X\right)$, which is then multiplied with masks ($M_{1}\cdot \theta_{e}\left(X\right),M_{2}\cdot \theta_{e}\left(X\right)$) for feature-wise perturbation. The regions of mask ($M_{1}$ or $M_{2}$) corresponding to 1 keep the feature of $\theta_{e}\left(X\right)$ unchanged, while the others are filled with 0. Figure 4b illustrates the workflow of Mask step at feature-level. Note that the features are visualized by GradCAM [36].

The Mix step draws on the idea of Mixup strategy and ensembles pseudo labels from the two segmentation branches. Figure 2 illustrates the workflow of the Mix step. Specifically, in this step, the outputs ($Y_{seg1}$, $Y_{seg2}$) of two segmentation branches are mixed according to the complementary masks ($M_{1},M_{2}$) and then integrated to generate pseudo labels ($Y_{pse}$), which are formulated as:(1)$$Y_{pse}=M_{1}\cdot Y_{seg1}+M_{2}\cdot Y_{seg2},$$
where $M_{1}\cdot Y_{seg1}$ returns 1 for masked regions in mask $M_{1}$ and 0 otherwise. The same is true for $M_{2}\cdot Y_{seg2}$. Consequently, the integration between the $Y_{seg1}$ and $Y_{seg2}$ is tight. Note that $M_{1}$ and $M_{2}$ change dynamically at each iteration, which enhances the generation of pseudo labels and prevents them from remembering themselves or not updating [37].

Consequently, scribbles annotations are enlarged by propagating the few labelled pixels in scribbles to vast unlabelled regions.

#### 3.2.3. Scribble-Supervised Coarsely Training

The segmentation branches are first coarsely trained by minimizing the dual-branch partial Cross-Entropy loss ($L_{DpCE}$), which is formulated as follows:(2)$$L_{DpCE}\left(Y_{seg1},Y_{seg2},Y\right)=-1\left(Y\right)\sum_{n=1}^{N}\left[\log\left(Y_{seg1}^{n}\right)+\log\left(Y_{seg2}^{n}\right)\right],$$
where *N* is the number of classes and *n* refers to the class of foreground. 1(Y) returns 1 for annotated pixels in the scribble *Y* and 0 otherwise. Only the gradients of annotated pixels are calculated for back propagation, and unlabelled regions are ignored.

#### 3.2.4. Self-Supervised Masked Image Modeling

Built upon the reconstruction branch in the siamese network, this section describes the self-supervised masked image modeling for optimizing dual-branch siamese network to extract features from sparse supervision.

Figure 2 illustrates the proposed MaskMix-based perturbation and reconstruction at the image-level and feature-level. For instance, at the image-level, the original image *X* is masked from perturbations of $M_{1},M_{2}$ in the Mask step (Section 3.2.2), and then the reconstruction branch ($\theta_{e}$ and $\theta d_{0}$) receives them and outputs image-like reconstructions $Y_{rec1},Y_{rec2}$, respectively. Above two predictions participants in the following self-supervised masked image modeling. The reconstruction loss ($L_{rec}$) is designed to carry out self-supervised masked image modeling, constraining the reconstruction results of dual branches to be consistent with each other. Meanwhile, these reconstruction results are also optimized the original image *X*.
(3)$$L_{rec}\left(Y_{rec1},Y_{rec2},X\right)={\left(Y_{rec1}-Y_{rec2}\right)}^{2}+{\left(Y_{rec1}-X\right)}^{2}+{\left(Y_{rec2}-X\right)}^{2}.$$

By minimizing the above reconstruction discrepancy via the self-supervised learning loss function $L_{rec}$, it is able to optimize the encoder θe and decoder $\theta d_{0}$ in conjunction with the proposed MaskMix strategy.

### 3.3. Pseudo Labels Optimization: Shape-Aware Adversarial Learning

Previous methods fail to locate the target boundary accurately due to the lack of target shape priors. Hence, this study incorporates boundary priors and co-trains the network by adversarial learning to regularize the target shape of prediction.

#### 3.3.1. Additional Unpaired Masks-Based Adversarial Learning Refinement

In order to regularize the generated pseudo labels to satisfy the shape supervision, this study adopts the idea of generative adversarial network [5]. An adversarial game is formed through the generation of pseudo labels and the discrimination between pseudo labels and real masks.

At the beginning, a discriminator $\theta_{dis}$ is constructed by a CNN-based decoder with a fully connected layer and receives an input <$Y_{pse}$,*Z*>, where $Y_{pse}$ denote the generated pseudo labels from siamese network via the Equation (Equation 1), and *Z* denote the real unpaired masks with the same ROI from additional dataset. Unpaired masks *Z* are incorporated as the additional priors from external datasets to regularize the shape of pseudo labels and improve segmentation performance. Specifically, unpaired masks *Z* (*real*) are fed to $\theta_{dis}$ for adversarial learning along with the pseudo labels $Y_{pse}$ (*fake*). The discriminate output is the bool prediction $Y_{dis}\in R^{B}$ that indicates that the category of input mask is fake or real.

The objective of shape-aware adversarial learning is to finely train the discriminator to optimize pseudo labels with shape priors. It formulates the adversarial objectives as an adversarial game, where the segmentation branches $\theta_{e},\theta d_{1},\theta d_{2}$ and discriminator $\theta_{dis}$ are jointly trained via an adversarial loss ($L_{adv}$):(4)$$L_{adv}\left(Y_{pse},Z\right)=E\left[\log\left(1-\theta_{dis}\left(Z\right)\right)\right]+E\left[\log\left(\theta_{dis}\left(Y_{pse}\right)\right)\right],$$
where $\theta_{dis}$ is trained to discriminate between *Z* and $Y_{pse}$ correctly and outputs the prediction $\theta_{dis}\left(Y_{pse}\right),\theta_{dis}\left(Z\right)$. The segmentation branches are trained to predict segmentation masks capable of tricking the discriminator and making their shapes satisfy the shape priors.

#### 3.3.2. MaskMix-Based Pseudo Label-Supervised Finely Training

After obtaining the pseudo labels in the Mix step and refining them via adversarial learning, this study further carries out pseudo label-supervised finely training.

Specifically, this section designs a Dice loss-based loss function $L_{pse}$ to optimize the training of $\theta_{e},\theta d_{1},\theta d_{2}$, where pseudo labels $Y_{pse}$ are used as fine-grained ground truth to calculate the discrepancy with respect to segmentation predictions ($Y_{seg1}$, $Y_{seg2}$). Pseudo label-supervised loss function $L_{pse}$ is formulated as follows:(5)$$L_{pse}\left(Y_{pse},Y_{seg1},Y_{seg2}\right)=1-\frac{\left|Y_{pse}\cap Y_{seg1}\right|}{\left|Y_{pse}|+|Y_{seg1}\right|}-\frac{\left|Y_{pse}\cap Y_{seg2}\right|}{\left|Y_{pse}|+|Y_{seg2}\right|}.$$

This design cuts off the gradient between segmentation decoders $\theta d_{1}$ and $\theta d_{2}$, not only maintaining their independence but also carrying out consistency learning.

### 3.4. Summary

The MaskMixAdv framework leverages shared shape priors from external datasets to enhance the performance of scribble-supervised medical image segmentation:During the training phase, the MaskMixAdv framework incorporates the MaskMix augmentation strategy into the DBSN network (θe, $\theta d_{0}$, $\theta d_{1}$, $\theta d_{2}$) to generate fine-grained pseudo labels. The shape-aware adversarial learning component regularizes the contours of these pseudo labels, facilitating model fine-tuning.During the testing phase, the reconstruction decoder $\theta d_{0}$ is no longer required, as its function is to facilitate training through self-supervised learning. The MaskMix strategy, which includes the mask step and the subsequent mix step, is not employed during inference because it serves to augment the sparse annotations specifically during the training phase. Considering that the output results and accuracy of the two segmentation decoders ($\theta d_{1}$ and $\theta d_{2}$) in the DBSN network are almost identical after adversarial scribble-supervised training, with no significant difference, the MaskMixAdv framework directly adopts the segmentation prediction result $Y_{seg1}$ from $\theta d_{1}$ as the final output. Additionally, the adversarial discriminator $\theta_{dis}$ is not utilized during inference, as its role is to refine pseudo labels during the training phase.

In a word, this framework serves as the foundation for scribble-supervised segmentation using MaskMixAdv on medical image datasets, including cardiac MRI segmentation in this study. MaskMixAdv enables the fine-grained interpretation of regions of interest (ROI) with accurate boundaries and reasonable shapes, improving the quality of segmentation results. For better understanding its mechanism, Appendix A releases the pseudocode of MaskMixAdv.

## 4. Case Study

The experiments conducted an assessment of the MaskMixAdv framework in the context of cardiac MRI segmentation. This specific task posed unique challenges within the broader field of medical image segmentation:Cardiac MRI images were affected by the constant motion of the heart, making it challenging to acquire clear images and perform accurate segmentation.The heart’s anatomy was complex, featuring multiple chambers, valves, and vascular structures closely packed together.Effective cardiac MRI segmentation should not only achieve high accuracy but also be efficient in terms of processing time to integrate seamlessly with clinical workflows.

Therefore, cardiac MRI segmentation served as an ideal scenario to assess the capabilities and practicality of the MaskMixAdv framework. This case study focused on the specific task of delineating the left ventricular cavity, myocardium, and right ventricle from MRI images. This segmentation task played a crucial role in clinical assessments, including the measurement of left and right ventricular ejection fractions, stroke volumes, left ventricle mass, and myocardium thickness. The case study conducted an investigation using two publicly available cardiac MRI datasets to achieve the following objectives:Compared the MaskMixAdv framework with several state-of-the-art scribble-supervised methods in terms of Dice and Hausdorff Distance (Section 4.2).Compared the MaskMix strategy with different masking augmentation techniques for scribble-supervised cardiac MRI segmentation (Section 4.2).Examined the sensitivity of MaskMixAdv to different combinations of scribble and mask annotations (Section 4.3).Validated the effectiveness of the shared shape prior extracted from other datasets for enhancing scribble-supervised cardiac MRI segmentation (Section 4.3).

### 4.1. Datasets and Settings

This subsection described the experimental protocols, including the (1) introduction of datasets, (2) evaluation metrics, (3) scribble annotation generation, (4) adversarial scribble-supervised segmentation setup, (5) pre-processing and data augmentations, and (6) network implementation and training setup.

**Introduction of datasets.** Two public cardiac MRI segmentation datasets were used in this section, including ACDC [38] and MSCMR [39,40], to evaluate the capabilities of MaskMixAdv. An overview of the two datasets was reported in Table 1. The ACDC dataset came from the MICCAI 2017 automated cardiac diagnosis challenge, which provided 200 short-axis cine-MRI scans from 100 patients. The MSCMR dataset came from the MICCAI 2019 multi-sequence cardiac MR segmentation challenge, which provided late gadolinium enhancement (LGE) cardiac MR images from 45 patients with cardiomyopathy. Both ACDC and MSCMR shared the same labelled cardiac ROI, i.e., right ventricle (RV), left ventricle (LV), and myocardium (Myo).

**Evaluation metrics.** During testing, only the prediction of $\theta d_{1}$ was regarded as the final output $Y_{seg1}$. Although the 3D volumes of source datasets were sliced into the 2D image during segmentation due to the large thickness, all 2D predictions were stacked in order slice by slice and integrated as a 3D volume for evaluation. The segmentation performance on each foreground class was measured separately in terms of 3D Dice (Implemented by package medpy.metric.binary.dc in this study.) and Hausdorff Distance (HD).
(6)$$3D\;Dice\left(Y_{seg1},Y_{GT}\right)=\frac{2|Y_{seg1}\cap Y_{GT}|}{\left|Y_{seg1}|+|Y_{GT}\right|}$$
(7)$$HD\left(Y_{seg1},Y_{GT}\right)=\max\left\{\max_{a\in Y_{seg1}}\min_{b\in Y_{GT}}d\left(a,b\right),\;\max_{b\in Y_{GT}}\min_{a\in Y_{seg1}}d\left(a,b\right)\right\},$$
where $Y_{GT}$ and $Y_{seg1}$ denote the dense pixel-wise annotations and predictions from the first segmentation branch, respectively. Moreover, $a\in Y_{seg1}$ represents any point *a* belongs to the set of contour points in $Y_{seg1}$, and d(a,b) indicates the distance between the two points *a* and *b*. Note that, in order to eliminate the impact of small outliers of cardiac targets, this study used the 95% Hausdorff Distance ($HD_{95}$), where the 95th percentile of the Euclidean distances between boundary points was calculated (Implemented by package medpy.metric.binary.hd95 in this study.). All results were evaluated without post-processing and reported in (mean ± std).

**Generation of scribble annotations.** The official ACDC and MSCMR datasets provided pixel-labelled mask annotations from medical experts without scribble annotations, which were the main supervision during the training in this study. Thus, in order to obtain scribble annotations of the two datasets, this study followed previous works [4,20,33] and used the same scribble annotations for fair comparisons. Specifically, Valvano et al. [41] generated the scribble annotations of the ACDC dataset, including all concerned cardiac structures and background, where the average image coverage of scribbles was: 0.1%, 0.2%, 0.1% and 10.4%, for RV, Myo, LV, and background, respectively. Following the same principles of [41], this study used the scribble annotations of the MSCMR dataset generated by [20,33].

**Adversarial scribble-supervised segmentation setup.** Since ACDC and MSCMR had the same target cardiac ROIs, i.e., RV, LV and Myo, there was a rationale behind the proposed adversarial scribble-supervised learning between the two datasets, with a view to optimizing the pseudo labels with the help of external masks. Here are two interrelated circumstances:When the source dataset was ACDC, the dataset was randomly divided into a training set and a test set in a 4:1 ratio, and then a five-fold cross-validation was completed. Apart from the paired image and scribble annotations from the source dataset (ACDC), dense annotations from the additional unpaired dataset (MSCMR) were used as unpaired masks *Z* for adversarial scribble-supervised segmentation.When the source dataset was MSCMR, this study adopted the same dataset split in ref. [20,33], i.e., 45 patients were divided into 25 cases as the training set, 5 cases as the validation set, and the remaining 15 cases as the test set. Apart from the paired image and scribble annotations from the source dataset (MSCMR), the unpaired masks *Z* were dense annotations from the additional unpaired dataset (ACDC).

**Pre-processing and data augmentations.** Following previous works [4,20,33], this study sliced the 3D volumes into the 2D images for segmentation. All 2D slices were normalized. Since the 2D slices from the ACDC and MSCMR datasets had different resolutions, this section first resampled all images to a fixed spacing of 1.37 × 1.37 mm and then central cropped to the size 212 × 212. Random rotation and flipping were then used for data augmentations.

**Network implementation and training setup.** The proposed dual-branch siamese network adopted the vanilla 2D U-Net [18] as the encoder ($\theta_{e}$)-decoder ($\theta d_{1}$) architecture, where two auxiliary decoders ($\theta d_{0},\theta d_{2}$) are extended for the reconstruction branch and segmentation branch, respectively. Note that all decoders ($\theta d_{0},\theta d_{1},\theta d_{2}$) share the same architecture but different weights and output channels. The SGD (weight decay = $1\times 10^{-4}$, momentum = 0.9) optimizer and Poly Learning Rate Policy [42] were applied during training. For fair comparisons, this study sets the same parameters as previous work [4] for the ACDC dataset, where the training iterations were 60,000 and batchsize was 12. For MSCMR dataset, this study followed previous works [20,33], where the training completed after 1000 epochs and batchsize was 16. The run-time infrastructure for the experiments was formed by PyTorch 1.10.0 with CUDA 10.2 over 4 NVIDIA 1080Ti GPUs, NVIDIA, Santa Clara, CA, USA.

### 4.2. Comparison with the State-of-the-Art Methods

In order to evaluate the advancement of the proposed MaskMixAdv framework, this section conducted comprehensive experiments on the two cardiac MRI segmentation datasets (ACDC and MSCMR), including: (1) compare MaskMixAdv with state-of-the-art scribble-supervised segmentation methods, and  (2) compare MaskMix with state-of-the-art data augmentation strategies at the image-level and feature-level.

Several state-of-the-art scribble-supervised segmentation methods were implemented in this study, including ScribbleSup [6], RandomWalks [9], USTM-Net [30], Sribble2Label [28], Gated CRF [43], MumfordLoss [44], EntropyMini [45], Regularized Loss [10], WSL4MIS [4], CycleMix [20], Puzzle Mix [24], ShapePU [33], CVIR [46], nnPU [47] and ZScribbleSeg [48].

Table 2 and Table 3 reported the results on the ACDC dataset and MSCMR dataset, respectively. Results indicated that the proposed MaskMixAdv outperformed the existing methods in most cardiac MRI segmentation tasks. Specifically, on the MSCMR dataset, MaskMixAdv achieved a 3D Dice improvement of 1.84 percentage points over the second-best method (ZScribbleSeg) and reduced $HD_{95}$ from 26.6 to 5.4. On the ACDC dataset, MaskMixAdv achieved a 3D Dice improvement of 1.72 percentage points over the second-best method (WSL4MIS) and reduced $HD_{95}$ from 6.9 to 4.7 on the ACDC dataset. Both of these advances were statistically significant with *p*-values < 0.05.

Figure 5 visualized the segmentation predictions of different methods. The results illustrated a large discrepancy between scribble-supervised baseline (i.e., $L_{DpCE}$) and mask-supervised method. But MaskMixAdv compensated it. Besides, compared with previous methods that generated misshapen predictions (e.g., WSL4MIS and ZScribbleSeg), MaskMixAdv generated more realistic and reasonable segmentation predictions. Above observations demonstrated the effectiveness of shape-aware adversarial learning that incorporated the boundary priors and encouraged the neural network to localize objects in the medical image. Above results also indicated the advancement of MaskMixAdv for promising performance in scribble-supervised cardiac MRI segmentation.

In order to evaluate the performance of the aforementioned methods after introducing adversarial learning, this study also conducted additional experiments on the source dataset ACDC, where the compared methods were re-implemented and integrated with the discriminator ($\theta_{dis}$). This study introduced the same unpaired 45 masks from additional unpaired dataset MSCMR for joint training with $L_{adv}$.

Figure 6 reported the results, which indicated that:If all methods incorporated these external masks, the proposed MaskMixAdv (0.887±0.051) outperformed the second-best one (WSL4MIS, 0.875±0.050) with statistical significance (*p*-value was 0.0182 < 0.05).If all methods did not use any external mask, the 3D Dice of MaskMixAdv (0.870±0.056) was still the-state-of-art and not statistically different from the best reported one (WSL4MIS, 0.872±0.077) with *p*-value of 0.7617 > 0.05. But MaskMixAdv still outperformed the re-implemented WSL4MIS (0.863±0.050).The proposed shape-aware adversarial learning (Section 3.3) was compatible to most scribble-supervised methods, whose performances improved after introducing external masks. Above observation demonstrated that it was possible to break the upper bound of scribble-supervised segmentation with the help of shape priors.

In addition, this study compared the proposed MaskMix augmentation strategy with several state-of-the-art data augmentation counterparts, including Mixup [19], Cutout [22], CutMix [23], Puzzle Mix [24], Co-mixup [21], CycleMix [20], Uniform Noise (Implemented by package torch.distribution.uniform() in this study.), Gaussian Noise (Implemented by package torch.distribution.normal() in this study.), Salt&Pepper Noise (Implemented by package skimage.util.random_noise(mode=’s&p’) in this study.), and Dropout (Implemented by package torch.nn.functional.dropout2d() in this study.). Figure 4 illustrated above augmentation strategies using cardiac MRI images as examples. For fair comparison, all methods were implemented with same experimental protocol, and the proposed MaskMix was incorporated at the image-level and feature-level, respectively.

Table 4 reported the results, which indicated that the proposed MaskMix strategy outperformed the existing augmentation methods in most metrics. Specifically, MaskMix outperformed the second-best method (Dropout) by 1.2 and 1.1 percentage points on 3D Dice. For Myo and LV segmentation, MaskMix outperformed the second-best method (Dropout) by 1.2 and 1.1 percentage points on 3D Dice, respectively. For RV segmentation, the best method (CycleMix) outperformed MaskMix by 0.5 percentage points on 3D Dice, and this discrepancy was not statistical different. Above results demonstrated the effectiveness and advancement of the MaskMix strategy in enhancing coarse-grained scribble annotations. Note that MaskMix obtained similar performance at both image-level and feature-level with no statistical difference, whereas the feature-level perturbation strategy was finally used in this study.

### 4.3. Ablation Study

Comprehensive ablation studies were conducted to evaluate the effectiveness of individual components in MaskMixAdv framework. Experiments compared: (1) the baseline model (vanilla U-Net trained by the loss $L_{pCE}$), (2) the proposed dual-branch siamese network with different scribble-supervised components in MaskMixAdv, i.e., $L_{DpCE},L_{rec},L_{pse},L_{adv}$. Besides, the data sensitivity of MaskMixAdv on different ratios of mask and scribble annotations was explored.

Table 5 reported the performance of each component of MaskMixAdv on the ACDC dataset and MSCMR dataset, respectively. Results indicated that:Scribble-supervised methods performed poorly when only $L_{pCE}$ and $L_{DpCE}$ were applied. In contrast, performance improved after introducing $L_{rec}$, demonstrating the effectiveness of the reconstruction branch in helping the dual-branch siamese network to extract features from sparse supervision.The performance of $L_{DpCE}+L_{rec}$ was still not satisfied but the pseudo labelling component ($L_{pse}$) improved it a lot, demonstrating the effectiveness of the pseudo labels in enhancing scribble-supervised segmentation.The $HD_{95}$ of $L_{DpCE}+L_{rec}+L_{pse}$ was still far below that of mask-supervised methods. This meant that predictions on some cardiac structures were incorrect with imprecise boundaries. However, the proposed adversarial learning ($L_{adv}$) optimized this drawback, reducing the $HD_{95}$ discrepancy between the scribble-supervised and mask-supervised methods.

Besides, semi-supervised experiments were conducted on the ACDC dataset to explore the data sensitivity of MaskMixAdv on different ratios of mask and scribble annotations.

First, this study explored the sensitivity of MaskMixAdv towards the number of scribbles, where different ratio of labelled data with scribble annotations instead of pixel-wise masks were used for training. Table 6 and Figure 7 reported the results. As can be observed:As expected, the performance of MaskMixAdv trended upwards as the percentage of labelled data increased.With decreasing amounts of labelled data, MaskMixAdv retained a high performance on cardiac MRI segmentation. Specifically, when only 50% of the labelled samples were accessible, MaskMixAdv could reach about 98% of the performance after training with all labelled samples.MaskMixAdv could also be trained on a few labelled datasets to achieve comparable performance to the full-labelled dataset. For instance, MaskMixAdv trained with 10% labelled data sacrificed only 12% of performance compared to training with all labelled data, but reduced the number of labelled samples by 90%, saving more on labelling costs.

Above results demonstrated that the proposed MaskMixAdv was insensitive to the number of scribbles and could achieve label-efficient adversarial scribble-supervised learning for cardiac MRI segmentation.

Second, this study also investigated the sensitivity of MaskMixAdv towards the number of external masks, where MaskMixAdv was trained under different number of masks. Figure 8 reported the results. It could be observed that:If only a small number of masks cases (<10 in this study) were incorporated, the optimization of pseudo labelling was rather harmful, as shown by the decrease of 3D Dice and the increase of Hausdorff distance.When a sufficient number of masks cases (≥10 in this study) were introduced, the performance of the model was rapidly improved, as shown by the increase of 3D Dice and the significant decrease of Hausdorff distance.

Above results demonstrated that the proposed MaskMixAdv was sensitive to the number of external unpaired masks. As a consequence, sufficient amount of same-ROI masks could be used as additional prior knowledge of the shape, contributing to improved adversarial scribble-supervised learning.

### 4.4. Discussions

This subsection discussed (1) the potential of applying MaskMixAdv to other coarse-grained annotations, (2) the labelling efficiency of scribble annotations, and (3) the advantages and limitations of MaskMixAdv.

#### 4.4.1. Extension to Point-Supervised Cardiac MRI Segmentation

Apart from scribbles annotations, there were other common coarse-grained annotations, such as bounding boxes [49], points [50], and image-level labels [51]. Among them, scribbles are the most flexible and user-friendly. In addition to scribbles, point annotation is another ideal coarse-grained weak annotation that provides labelling information for only a number of discrete points, having much less annotation cost. Point annotation can be considered an extreme case of scribble annotation with less supervision but more unknown regions. The comparison between the two is shown in the Figure 9.

Table 7 reported the performance of MaskMixAdv extended for point-supervised cardiac MRI segmentation on the ACDC dataset. Results demonstrated the effectiveness of the proposed MaskMixAdv when extended to point-supervision. Specifically, when only 1 point annotation of each class was accessible, MaskMixAdv could reach about 80% of the performance of scribble-supervised training, and about 78% of the performance of mask-supervised training. Besides, the performance gap between 3 points-supervised and scribbles-supervised MaskMixAdv was further narrowed, which demonstrated the scalability of this work.

#### 4.4.2. Efficiency of Scribble-Based Cardiac MRI Annotations

On the one hand, this study also investigated the time cost of mask and scribble annotations on cardiac MRI images. The conclusion was that, compared to fine-grained mask annotations that took about 20 min per case (https://zmiclab.github.io/zxh/0/mscmrseg19/data.html accessed on 1 January 2024), coarse-grained scribble annotations were measured in seconds, thus saving at least 95% of the annotation time.

On the other hand, Table 6 and Table 7 also indicated that this work narrowed the performance gap between mask-supervised and scribble-supervised segmentation methods with label-efficiency, i.e., sacrificing only 2.6 percentage points of 3D Dice on the ACDC dataset for less labelling time.

#### 4.4.3. Advantages and Limitations

Overall, MaskMixAdv made significant progresses towards solving the open issues (Section 1): (1) Dense pixel-wise prediction could be obtained from scribble-labelled MRI images by generating pseudo labels via MaskMix-based scribble-supervised learning siamese network, and (2) Boundary information about the object structure had been incorporated via shape-aware adversarial learning, and misshapen pseudo labels were optimized with shape priors.

However, there were still some limitations of MaskMixAdv to be further explored:The MaskMixAdv proposed in this manuscript was designed for 2D MRI images with 2D scribbles. Therefore, the direct application of the current MaskMixAdv to 3D volumes with 3D scribbles or other coarse-grained annotations for weakly-supervised learning was not validated.The external masks for MaskMixAdv were from the same modality as source dataset, i.e., the ACDC and MSCMR in this study. So the superiority of this method to perform adversarial scribble-supervised learning on cross-modality medical images dataset remained to be validated.

## 5. Conclusions

In order to benefit from the merits of scribble annotations while approaching the level of precision achieved by fine-grained annotations in medical image segmentation, this study developed a shape-aware scribble-supervised learning framework (MaskMixAdv). The MaskMixAdv framework successfully accomplished its goal through two main tasks:Pseudo label generation: This task was facilitated by a dual-branch siamese network equipped with the MaskMix augmentation strategy. The siamese network, utilizing MaskMix-based augmentation, enriched the coarse-grained annotations through both reconstruction and segmentation branches, effectively enhancing the quality of the scribble annotations.Pseudo label optimization: In this task, adversarial learning was employed to refine the generated pseudo labels by transferring shape priors from additional datasets. This process aimed to ensure that the pseudo labels accurately matched real shapes.

In a case study focused on cardiac MRI segmentation, the MaskMixAdv framework was evaluated for its effectiveness in utilizing shared shape priors. Compared to state-of-the-art scribble-supervised methods like ZScribbleSeg and WSL4MIS, MaskMixAdv demonstrated its superiority with an average 3D Dice value of 0.887 and an average Hausdorff distance of 4.7 mm, outperforming its counterparts with statistical significance. Additionally, MaskMixAdv narrowed the performance gap between scribble-supervised and mask-supervised methods, reducing annotation time by at least 95% while only sacrificing 2.6 percentage points of Dice performance. Moreover, MaskMixAdv exhibited strong generalization capabilities to point-supervised annotations, making it a viable alternative to other Mixup-based cross-sample data augmentation methods.

## Figures and Tables

**Figure 1 bioengineering-11-01146-f001:**
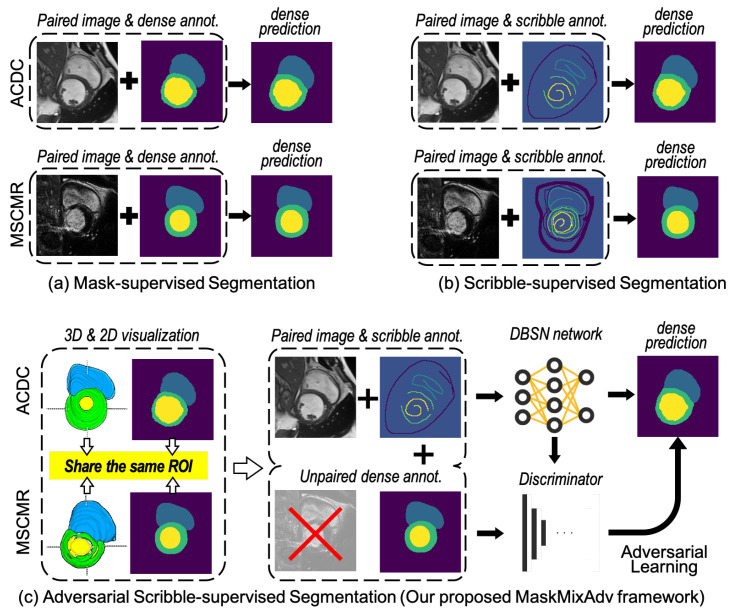
An overview of medical image segmentation under different three kinds of supervisions: (**a**) mask-supervised segmentation based on paired images and pixel-labelled masks, (**b**) scribble-supervised segmentation based on paired images and coarse-grained scribbles annotations, and (**c**) adversarial scribble-supervised segmentation based on paired images, coarse-grained scribbles, and additional unpaired masks. Cases from two cardiac MRI datasets (ACDC and MSCMR) are shown to give a conceptual comparison. As can be observed, since the regions of interest (ROI) among ACDC and MSCMR datasets are shared, it is reasonable to transfer shape prior across the two datasets.

**Figure 2 bioengineering-11-01146-f002:**
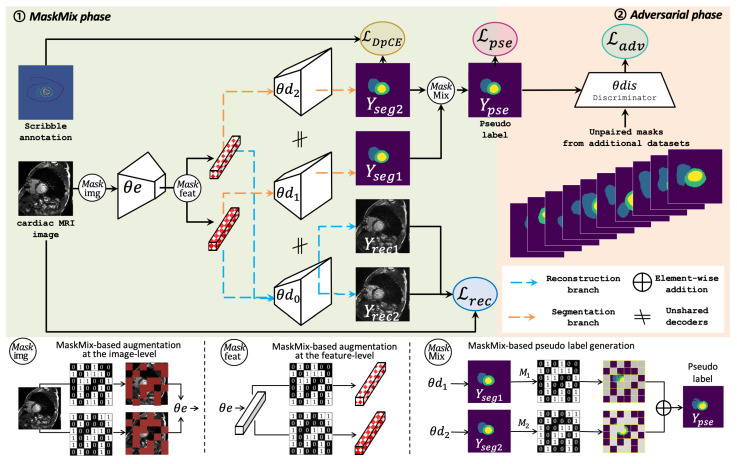
The backbone of the proposed MaskMixAdv framework is a dual-branch siamese network (DBSN), and a CNN-based discriminator ($\theta_{dis}$) is built on top of the backbone for adversarial learning. MaskMixAdv consists of two phases, where the first phase (MaskMix) performs data augmentation and scribble-supervised learning, the second phase (Adv) achieves adversarial learning. Note that some connecting lines of the loss function in the figure are omitted for better observation.

**Figure 3 bioengineering-11-01146-f003:**
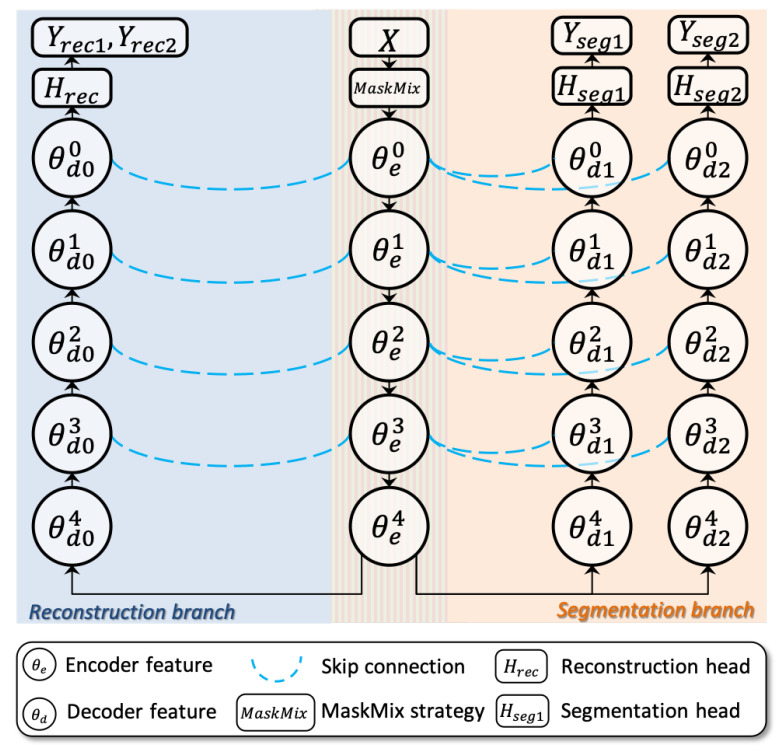
The architecture of the proposed siamese network DBSN.

**Figure 4 bioengineering-11-01146-f004:**
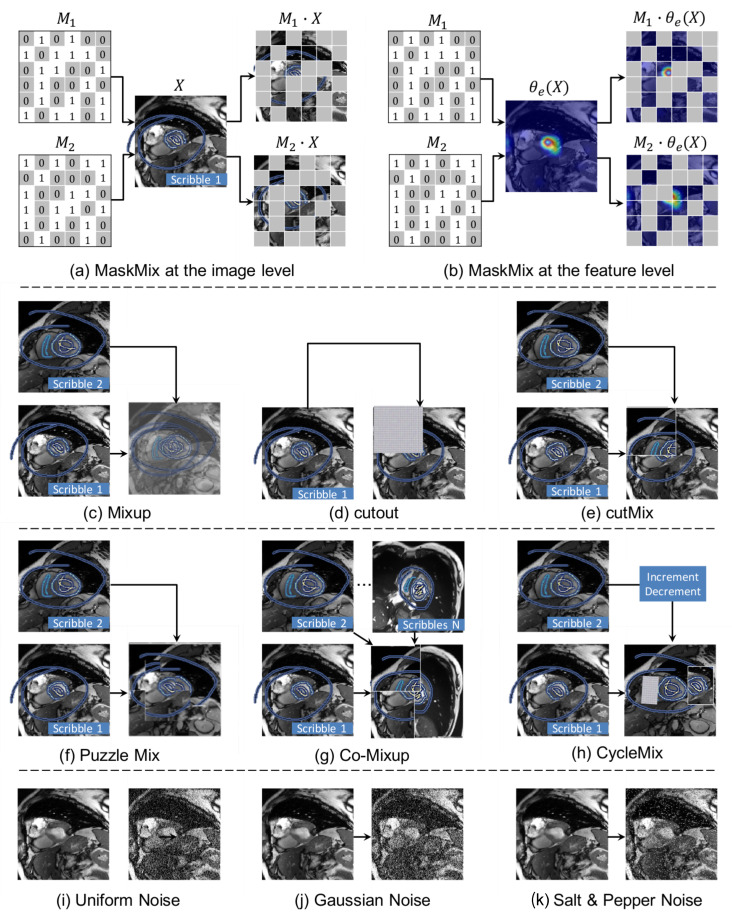
Illustration of Mask phase in the proposed MaskMixAdv framework and other methods for data augmentation. For the Mixup-based approaches, this figure introduces white outlines to easily distinguish the multi-sample mixing process. Only perturbations at the image level are shown here, and perturbations at the feature level are similar and thus omitted. Note the scribbles shown here are bolded for ease of viewing. Better zoom in for more details.

**Figure 5 bioengineering-11-01146-f005:**
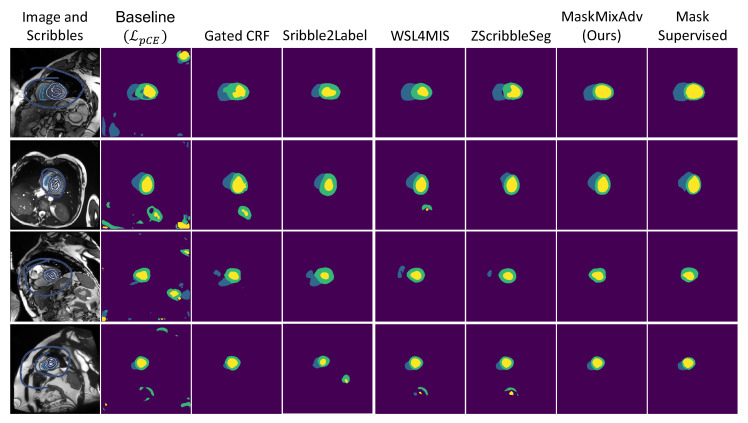
Visualization of the results of the proposed MaskMixAdv and other methods for cardiac MRI segmentation on ACDC dataset. Note that the scribbles shown are bolded for ease of viewing.

**Figure 6 bioengineering-11-01146-f006:**
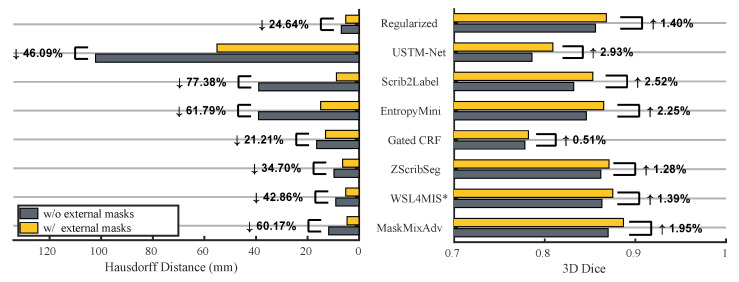
Comparison with existing scribble-supervised segmentation methods with and without external masks on the ACDC dataset. ↑ and ↓ denote the metrics improved and reduced after incorporating external masks, respectively. * The performance of WSL4MIS implemented by this study is evaluated.

**Figure 7 bioengineering-11-01146-f007:**
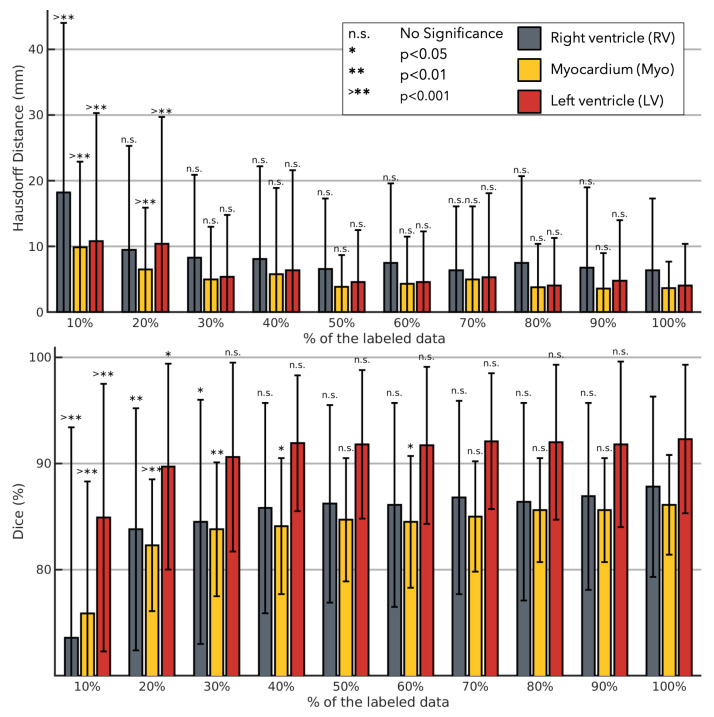
Results of the semi-supervised learning experiments under different label fractions. The gray, orange, and red blocks indicated the performance (HD95 and Dice) of right ventricle (RV), myocardium(Myo), and left ventricle (LV) by MaskMixAdv, respectively. In addition, statistical significance analysis is conducted on a case-by-case basis between the 100% labelled results and the results labelled from 10% to 90%, whose *p*-values are reported as * or n.s.

**Figure 8 bioengineering-11-01146-f008:**
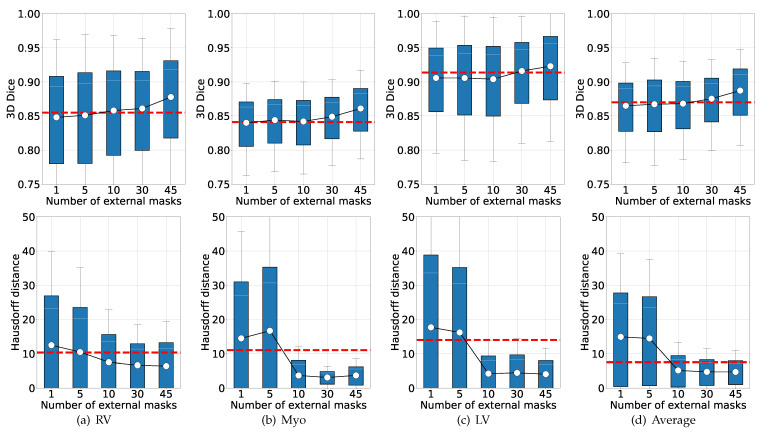
Box illustration of the performance of MaskMixAdv with different number of masks from the additional unpaired dataset. From (**a**–**d**), the results presented RV, Myo, LV, and the average value in order. The first row reported the 3D Dice results, while the second row reported the Hausdorff Distance. Note that the white circles denoted the mean values. The dotted red lines indicated the performance of proposed MaskMix, which was trained without $L_{adv}$, i.e., the number of external masks cases was 0.

**Figure 9 bioengineering-11-01146-f009:**
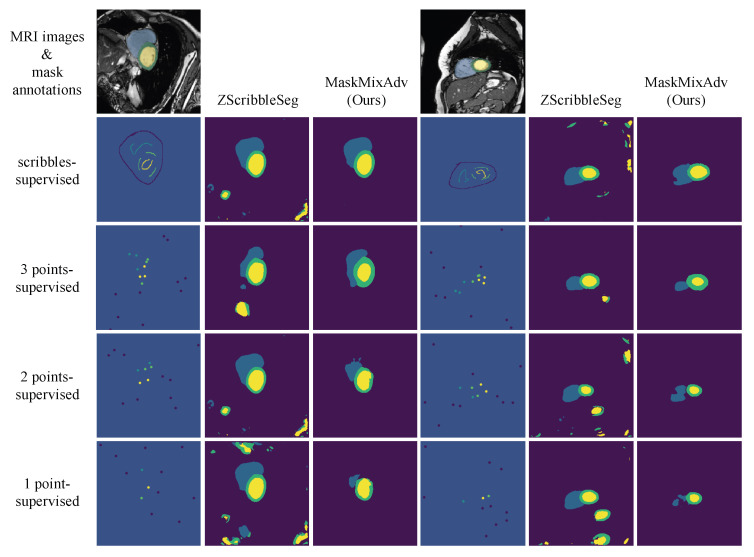
Comparison between different annotations, including fine-grained masks, coarse-grained scribbles, and coarse-grained points. The points shown here are bolded for ease of viewing. Note that the above three contain supervisory information in descending order.

**Table 1 bioengineering-11-01146-t001:** An overview of the datasets used in this section. These MRI datasets share the same target structure, i.e., right ventricle (RV), myocardium (Myo), and left ventricle (LV).

Abbr.	Dataset Source	Modality	Input Size	Patients
ACDC	Automated Cardiac Diagnosis Challenge	MRI	212 × 212	100
MSCMR	Multi-sequence Cardiac MR Segmentation Challenge	MRI	224 × 224	45

**Table 2 bioengineering-11-01146-t002:** Comparison with existing scribble-supervised methods on the ACDC dataset. All results are based on the *5-fold cross-validation* with the same backbone (U-Net). Results style: **best**, second-best, /denoted not reported in the original paper.

Source Dataset: ACDC, Additional Unpaired Dataset: MSCMR								
Methods	*3D Dice* ↑				${\mathrm{HD}}_{95}↓$			
	RV	Myo	LV	Average	RV	Myo	LV	Average
ScribbleSup [6]	$0.625_{\;\pm \;0.160}$	$0.668_{\;\pm \;0.095}$	$0.766_{\;\pm \;0.156}$	$0.686_{\;\pm \;0.137}$	$187.2_{\;\pm \;35.2}$	$165.1_{\;\pm \;34.4}$	$167.7_{\;\pm \;55.0}$	$173.3_{\;\pm \;41.5}$
RandomWalker [9]	$0.813_{\;\pm \;0.113}$	$0.708_{\;\pm \;0.066}$	$0.844_{\;\pm \;0.091}$	$0.788_{\;\pm \;0.090}$	$11.1_{\;\pm \;17.3}$	$9.8_{\;\pm \;8.9}$	$9.2_{\;\pm \;13.0}$	$10.0_{\;\pm \;13.1}$
USTM-Net [30]	$0.815_{\;\pm \;0.115}$	$0.756_{\;\pm \;0.081}$	$0.785_{\;\pm \;0.162}$	$0.786_{\;\pm \;0.119}$	$54.7_{\;\pm \;65.7}$	$112.2_{\;\pm \;54.1}$	$139.6_{\;\pm \;57.7}$	$102.2_{\;\pm \;59.2}$
Sribble2Label [28]	$0.833_{\;\pm \;0.103}$	$0.806_{\;\pm \;0.069}$	$0.856_{\;\pm \;0.121}$	$0.832_{\;\pm \;0.098}$	$14.6_{\;\pm \;30.9}$	$37.1_{\;\pm \;49.4}$	$65.2_{\;\pm \;65.1}$	$38.9_{\;\pm \;48.5}$
Gated CRF [43]	$0.743_{\;\pm \;0.164}$	$0.729_{\;\pm \;0.134}$	$0.862_{\;\pm \;0.125}$	$0.778_{\;\pm \;0.125}$	$18.7_{\;\pm \;21.8}$	$13.8_{\;\pm \;19.7}$	$16.9_{\;\pm \;26.4}$	$16.5_{\;\pm \;19.3}$
MumfordLoss [44]	$0.809_{\;\pm \;0.093}$	$0.832_{\;\pm \;0.055}$	$0.876_{\;\pm \;0.093}$	$0.839_{\;\pm \;0.080}$	$17.1_{\;\pm \;30.8}$	$28.2_{\;\pm \;43.2}$	$37.9_{\;\pm \;59.6}$	$27.7_{\;\pm \;44.5}$
EntropyMini [45]	$0.839_{\;\pm \;0.108}$	$0.812_{\;\pm \;0.062}$	$0.887_{\;\pm \;0.099}$	$0.846_{\;\pm \;0.089}$	$25.7_{\;\pm \;44.5}$	$47.4_{\;\pm \;50.6}$	$43.8_{\;\pm \;57.6}$	$39.0_{\;\pm \;50.9}$
Regularized [10]	$0.856_{\;\pm \;0.101}$	$0.817_{\;\pm \;0.054}$	$0.896_{\;\pm \;0.086}$	$0.856_{\;\pm \;0.080}$	$7.9_{\;\pm \;12.6}$	$6.0_{\;\pm \;6.9}$	$7.0_{\;\pm \;13.5}$	$6.9_{\;\pm \;11.0}$
Puzzle Mix [24]	$0.806_{\;\pm \;0.162}$	$0.715_{\;\pm \;0.125}$	$0.821_{\;\pm \;0.158}$	$0.781_{\;\pm \;0.148}$	/	/	/	/
WSL4MIS [4] (reported)	$0.861_{\;\pm \;0.096}$	$0.842_{\;\pm \;0.054}$	$0.913_{\;\pm \;0.082}$	$0.872_{\;\pm \;0.077}$	$7.9_{\;\pm \;12.5}$	$9.7_{\;\pm \;23.2}$	$12.1_{\;\pm \;27.2}$	$9.9_{\;\pm \;21.0}$
WSL4MIS [4] (implemented)	$0.848_{\;\pm \;0.091}$	$0.837_{\;\pm \;0.051}$	$0.904_{\;\pm \;0.065}$	$0.863_{\;\pm \;0.050}$	$9.0_{\;\pm \;17.1}$	$9.4_{\;\pm \;21.4}$	$8.7_{\;\pm \;21.2}$	$9.1_{\;\pm \;15.8}$
ShapePU [33]	$0.854_{\;\pm \;0.089}$	$0.813_{\;\pm \;0.095}$	$0.888_{\;\pm \;0.103}$	$0.851_{\;\pm \;0.096}$	/	/	/	/
CycleMix [20]	$0.860_{\;\pm \;0.089}$	$0.825_{\;\pm \;0.072}$	$0.880_{\;\pm \;0.115}$	$0.855_{\;\pm \;0.092}$	/	/	/	/
ZScribbleSeg [48]	$0.900_{\;\pm \;0.065}$	$0.825_{\;\pm \;0.069}$	$0.862_{\;\pm \;0.102}$	$0.862_{\;\pm \;0.086}$	$7.7_{\;\pm \;6.9}$	$8.9_{\;\pm \;6.4}$	$12.7_{\;\pm \;12.5}$	$9.8_{\;\pm \;9.2}$
MaskMix (Ours w/o $L_{adv}$)	$0.855_{\;\pm \;0.092}$	$0.841_{\;\pm \;0.055}$	$0.914_{\;\pm \;0.056}$	$0.870_{\;\pm \;0.056}$	$10.4_{\;\pm \;20.6}$	$11.1_{\;\pm \;27.3}$	$14.0_{\;\pm \;34.4}$	$11.8_{\;\pm \;21.8}$
MaskMixAdv (Ours w/ $L_{adv}$)	$0.878_{\;\pm \;0.085}$	$0.861_{\;\pm \;0.047}^{\;★}$	$0.923_{\;\pm \;0.070}$	$0.887_{\;\pm \;0.051}^{\;★}$	$6.4_{\;\pm \;10.9}$	$3.7_{\;\pm \;4.0}^{\;★}$	$4.1_{\;\pm \;6.3}^{\;★}$	$4.7_{\;\pm \;5.2}^{\;★}$

★: Statistical analysis between the second-best method (WSL4MIS) and MaskMixAdv was also conducted, where ^★^ denoted MaskMixAdv significantly outperformed WSL4MIS with *p*-value < 0.05.

**Table 3 bioengineering-11-01146-t003:** Comparison with existing scribble-supervised methods on the MSCMR dataset. Results style: **best**, second-best.

Source Dataset: MSCMR, Additional Unpaired Dataset: ACDC								
Methods	*3D Dice* ↑				${\mathrm{HD}}_{95}↓$			
	RV	Myo	LV	Average	RV	Myo	LV	Average
CVIR [46]	$0.404_{\;\pm \;0.110}$	$0.371_{\;\pm \;0.088}$	$0.331_{\;\pm \;0.076}$	$0.368_{\;\pm \;0.095}$	$180.9_{\;\pm \;55.4}$	$243.0_{\;\pm \;13.8}$	$259.2_{\;\pm \;14.2}$	$227.7_{\;\pm \;47.6}$
nnPU [47]	$0.432_{\;\pm \;0.100}$	$0.538_{\;\pm \;0.081}$	$0.341_{\;\pm \;0.067}$	$0.437_{\;\pm \;0.115}$	$199.7_{\;\pm \;57.5}$	$201.6_{\;\pm \;67.0}$	$259.4_{\;\pm \;14.2}$	$220.2_{\;\pm \;57.7}$
ShapePU [33]	$0.833_{\;\pm \;0.087}$	$0.785_{\;\pm \;0.080}$	$0.880_{\;\pm \;0.046}$	$0.833_{\;\pm \;0.082}$	$178.1_{\;\pm \;25.4}$	$189.4_{\;\pm \;55.8}$	$178.0_{\;\pm \;50.9}$	$181.8_{\;\pm \;45.3}$
CycleMix [20]	$0.835_{\;\pm \;0.041}$	$0.730_{\;\pm \;0.047}$	$0.748_{\;\pm \;0.064}$	$0.771_{\;\pm \;0.069}$	$73.4_{\;\pm \;51.4}$	$28.3_{\;\pm \;20.8}$	$224.6_{\;\pm \;35.3}$	$108.7_{\;\pm \;92.7}$
WSL4MIS [4]	$0.828_{\;\pm \;0.101}$	$0.815_{\;\pm \;0.033}$	$0.902_{\;\pm \;0.040}$	$0.848_{\;\pm \;0.076}$	$32.1_{\;\pm \;6.6}$	$42.1_{\;\pm \;13.5}$	$56.0_{\;\pm \;4.9}$	$43.4_{\;\pm \;31.0}$
Gated CRF [43]	$0.848_{\;\pm \;0.073}$	$0.825_{\;\pm \;0.032}$	$0.917_{\;\pm \;0.044}$	$0.863_{\;\pm \;0.066}$	$32.8_{\;\pm \;5.6}$	$37.9_{\;\pm \;5.1}$	$25.7_{\;\pm \;4.4}$	$32.2_{\;\pm \;7.1}$
ZScribbleSeg [48]	$0.854_{\;\pm \;0.055}$	$0.834_{\;\pm \;0.039}$	$0.922_{\;\pm \;0.039}$	$0.870_{\;\pm \;0.058}$	$51.0_{\;\pm \;39.3}$	$16.5_{\;\pm \;19.1}$	$12.1_{\;\pm \;14.7}$	$26.6_{\;\pm \;31.4}$
MaskMix (Ours w/o $L_{adv}$)	$0.877_{\;\pm \;0.059}$	$0.826_{\;\pm \;0.048}$	$0.909_{\;\pm \;0.051}$	$0.871_{\;\pm \;0.043}$	$39.3_{\;\pm \;55.3}$	$7.3_{\;\pm \;8.4}$	$27.5_{\;\pm \;39.3}$	$24.7_{\;\pm \;23.9}$
MaskMixAdv (Ours w/ $L_{adv}$)	$0.882_{\;\pm \;0.049}$	$0.847_{\;\pm \;0.035}$	$0.928_{\;\pm \;0.033}$	$0.886_{\;\pm \;0.032}$	$5.8_{\;\pm \;3.1}$	$4.9_{\;\pm \;3.1}$	$5.5_{\;\pm \;5.1}$	$5.4_{\;\pm \;3.2}$

**Table 4 bioengineering-11-01146-t004:** Performance of scribble-supervised methods on cardiac MRI segmentation with different data augmentation strategies. Results style: **best**, second-best.

Source Dataset: ACDC, Additional Unpaired Dataset: MSCMR				
Method	*3D Dice*			
	RV	Myo	LV	Average
Baseline ^0^	$0.525_{\;\pm \;0.184}$	$0.512_{\;\pm \;0.103}$	$0.704_{\;\pm \;0.159}$	$0.581_{\;\pm \;0.107}$
Mixup	$0.774_{\;\pm \;0.158}$	$0.715_{\;\pm \;0.129}$	$0.804_{\;\pm \;0.152}$	$0.765_{\;\pm \;0.146}$
Cutout	$0.817_{\;\pm \;0.123}$	$0.758_{\;\pm \;0.134}$	$0.815_{\;\pm \;0.172}$	$0.797_{\;\pm \;0.143}$
CutMix	$0.784_{\;\pm \;0.197}$	$0.726_{\;\pm \;0.176}$	$0.767_{\;\pm \;0.237}$	$0.759_{\;\pm \;0.203}$
Puzzle Mix	$0.806_{\;\pm \;0.162}$	$0.715_{\;\pm \;0.125}$	$0.821_{\;\pm \;0.15}$	$0.781_{\;\pm \;0.148}$
Co-mixup	$0.803_{\;\pm \;0.097}$	$0.689_{\;\pm \;0.153}$	$0.709_{\;\pm \;0.23}$	$0.734_{\;\pm \;0.161}$
CycleMix	$0.860_{\;\pm \;0.089}$	$0.825_{\;\pm \;0.072}$	$0.880_{\;\pm \;0.115}$	$0.855_{\;\pm \;0.092}$
Uniform Noise	$0.842_{\;\pm \;0.098}$	$0.778_{\;\pm \;0.111}$	$0.826_{\;\pm \;0.152}$	$0.816_{\;\pm \;0.080}$
Gaussian Noise	$0.692_{\;\pm \;0.278}$	$0.672_{\;\pm \;0.280}$	$0.781_{\;\pm \;0.252}$	$0.715_{\;\pm \;0.262}$
Salt&Pepper Noise	$0.839_{\;\pm \;0.107}$	$0.814_{\;\pm \;0.064}$	$0.903_{\;\pm \;0.088}$	$0.852_{\;\pm \;0.066}$
Image-level MaskMix ^1^	$0.847_{\;\pm \;0.101}$	$0.844_{\;\pm \;0.049}$	$0.911_{\;\pm \;0.074}$	$0.867_{\;\pm \;0.055}$
Dropout	$0.858_{\;\pm \;0.096}$	$0.831_{\;\pm \;0.058}$	$0.904_{\;\pm \;0.086}$	$0.864_{\;\pm \;0.061}$
Uniform Noise	$0.792_{\;\pm \;0.167}$	$0.778_{\;\pm \;0.131}$	$0.855_{\;\pm \;0.117}$	$0.808_{\;\pm \;0.120}$
Gauss Noise	$0.717_{\;\pm \;0.222}$	$0.709_{\;\pm \;0.169}$	$0.859_{\;\pm \;0.098}$	$0.762_{\;\pm \;0.139}$
Salt&Pepper Noise	$0.835_{\;\pm \;0.109}$	$0.781_{\;\pm \;0.143}$	$0.878_{\;\pm \;0.097}$	$0.832_{\;\pm \;0.096}$
Feature-level MaskMix ^1^	$0.855_{\;\pm \;0.092}$	$0.841_{\;\pm \;0.055}$	$0.914_{\;\pm \;0.073}$	$0.870_{\;\pm \;0.056}$

^0^ Implemented by the basic dual-branch siamese network with $L_{DpCE}$. ^1^ MaskMix denotes the proposed MaskMixAdv trained at the image-level or feature-level perturbations without adversarial learning (w/o $L_{adv}$) for fair comparisons.

**Table 5 bioengineering-11-01146-t005:** Performance of the proposed adversarial scribble-supervised learning components on ACDC and MSCMR datasets.

Source Dataset: ACDC, Additional Unpaired Dataset: MSCMR										
**Method**	$L_{\mathrm{pCE}}$	$L_{\mathrm{DpCE}}$	$L_{\mathrm{rec}}$	$L_{\mathrm{pse}}$	$L_{\mathrm{adv}}$	*** 3D Dice*** ↑				${\mathrm{HD}}_{95}$ ↓
						**RV**	**Myo**	**LV**	**Average**	**Average**
Baseline	✓					$0.462_{\;\pm \;0.176}$	$0.446_{\;\pm \;0.086}$	$0.556_{\;\pm \;0.182}$	$0.488_{\;\pm \;0.112}$	$184.9_{\;\pm \;23.5}$
MaskMixAdv		✓				$0.525_{\;\pm \;0.184}$	$0.512_{\;\pm \;0.103}$	$0.704_{\;\pm \;0.159}$	$0.581_{\;\pm \;0.107}$	$188.0_{\;\pm \;28.7}$
		✓	✓			$0.679_{\;\pm \;0.158}$	$0.620_{\;\pm \;0.094}$	$0.764_{\;\pm \;0.153}$	$0.688_{\;\pm \;0.098}$	$178.3_{\;\pm \;49.2}$
		✓	✓	✓		$0.855_{\;\pm \;0.092}$	$0.841_{\;\pm \;0.055}$	$0.914_{\;\pm \;0.073}$	$0.870_{\;\pm \;0.056}$	$11.8_{\;\pm \;21.8}$
		✓	✓	✓	✓	$0.878_{\;\pm \;0.085}$	$0.861_{\;\pm \;0.047}$	$0.923_{\;\pm \;0.070}$	$0.887_{\;\pm \;0.051}$	$4.7_{\;\pm \;5.2}$
** Source Dataset: MSCMR, Additional Unpaired Dataset: ACDC**										
**Method**	$L_{\mathrm{pCE}}$	$L_{\mathrm{DpCE}}$	$L_{\mathrm{rec}}$	$L_{\mathrm{pse}}$	$L_{\mathrm{adv}}$	***3D Dice*** ↑				${\mathrm{HD}}_{95}$ ↓
						**RV**	**Myo**	**LV**	**Average**	**Average**
Baseline	✓					$0.058_{\;\pm \;0.023}$	$0.582_{\;\pm \;0.067}$	$0.514_{\;\pm \;0.078}$	$0.385_{\;\pm \;0.243}$	$248.3_{\;\pm \;21.6}$
MaskMixAdv		✓				$0.715_{\;\pm \;0.096}$	$0.521_{\;\pm \;0.081}$	$0.573_{\;\pm \;0.119}$	$0.603_{\;\pm \;0.082}$	$202.6_{\;\pm \;13.9}$
		✓	✓			$0.634_{\;\pm \;0.201}$	$0.596_{\;\pm \;0.106}$	$0.744_{\;\pm \;0.106}$	$0.658_{\;\pm \;0.109}$	$155.4_{\;\pm \;25.7}$
		✓	✓	✓		$0.877_{\;\pm \;0.059}$	$0.826_{\;\pm \;0.048}$	$0.909_{\;\pm \;0.051}$	$0.871_{\;\pm \;0.043}$	$24.7_{\;\pm \;23.9}$
		✓	✓	✓	✓	$0.882_{\;\pm \;0.049}$	$0.847_{\;\pm \;0.035}$	$0.928_{\;\pm \;0.033}$	$0.886_{\;\pm \;0.032}$	$5.4_{\;\pm \;3.2}$

**Table 6 bioengineering-11-01146-t006:** Data sensitivity study: the performance of MaskMixAdv with different ratio of scribbles without pixel-wise full annotations.

Label Ratio	Num of Scribbles	*3D Dice* ↑				${\mathrm{HD}}_{95}$ ↓			
		RV	Myo	LV	Average	RV	Myo	LV	Average
10%	8	$0.736_{\;\pm \;0.198}$	$0.759_{\;\pm \;0.124}$	$0.849_{\;\pm \;0.126}$	$0.782_{\;\pm \;0.138}$	$18.2_{\;\pm \;25.8}$	$9.9_{\;\pm \;13.0}$	$10.8_{\;\pm \;19.5}$	$13.0_{\;\pm \;14.3}$
20%	16	$0.838_{\;\pm \;0.114}$	$0.823_{\;\pm \;0.062}$	$0.897_{\;\pm \;0.097}$	$0.853_{\;\pm \;0.073}$	$9.5_{\;\pm \;15.8}$	$6.5_{\;\pm \;9.4}$	$10.4_{\;\pm \;19.3}$	$8.8_{\;\pm \;11.4}$
30%	24	$0.845_{\;\pm \;0.115}$	$0.838_{\;\pm \;0.063}$	$0.906_{\;\pm \;0.089}$	$0.863_{\;\pm \;0.078}$	$8.3_{\;\pm \;12.6}$	$5.0_{\;\pm \;8.0}$	$5.4_{\;\pm \;9.4}$	$6.2_{\;\pm \;7.5}$
40%	32	$0.858_{\;\pm \;0.099}$	$0.841_{\;\pm \;0.064}$	$0.919_{\;\pm \;0.064}$	$0.872_{\;\pm \;0.061}$	$8.1_{\;\pm \;14.1}$	$5.8_{\;\pm \;13.1}$	$6.4_{\;\pm \;15.2}$	$6.8_{\;\pm \;10.9}$
50%	40	$0.862_{\;\pm \;0.093}$	$0.847_{\;\pm \;0.058}$	$0.918_{\;\pm \;0.070}$	$0.877_{\;\pm \;0.058}$	$6.6_{\;\pm \;10.7}$	$3.9_{\;\pm \;4.8}$	$4.6_{\;\pm \;7.9}$	$5.0_{\;\pm \;5.8}$
60%	48	$0.861_{\;\pm \;0.096}$	$0.845_{\;\pm \;0.062}$	$0.917_{\;\pm \;0.074}$	$0.874_{\;\pm \;0.061}$	$7.5_{\;\pm \;12.1}$	$4.3_{\;\pm \;7.2}$	$4.6_{\;\pm \;7.7}$	$5.5_{\;\pm \;6.8}$
70%	56	$0.868_{\;\pm \;0.091}$	$0.850_{\;\pm \;0.052}$	$0.921_{\;\pm \;0.064}$	$0.879_{\;\pm \;0.053}$	$6.4_{\;\pm \;9.7}$	$5.0_{\;\pm \;11.1}$	$5.3_{\;\pm \;12.8}$	$5.5_{\;\pm \;8.9}$
80%	64	$0.864_{\;\pm \;0.093}$	$0.856_{\;\pm \;0.049}$	$0.920_{\;\pm \;0.073}$	$0.880_{\;\pm \;0.055}$	$7.5_{\;\pm \;13.2}$	$3.8_{\;\pm \;6.6}$	$4.1_{\;\pm \;7.2}$	$5.2_{\;\pm \;7.3}$
90%	72	$0.869_{\;\pm \;0.088}$	$0.856_{\;\pm \;0.049}$	$0.918_{\;\pm \;0.078}$	$0.882_{\;\pm \;0.055}$	$6.8_{\;\pm \;12.2}$	$3.6_{\;\pm \;5.4}$	$4.8_{\;\pm \;9.2}$	$5.1_{\;\pm \;6.5}$
100%	80	$0.878_{\;\pm \;0.085}$	$0.861_{\;\pm \;0.047}$	$0.923_{\;\pm \;0.070}$	$0.887_{\;\pm \;0.051}$	$6.4_{\;\pm \;10.9}$	$3.7_{\;\pm \;4.0}$	$4.1_{\;\pm \;6.3}$	$4.7_{\;\pm \;5.2}$

**Table 7 bioengineering-11-01146-t007:** MaskMixAdv was extended to point-supervised cardiac MRI segmentation on the ACDC dataset. The minor performance gap demonstrated the scalability of this work.

Supervision	*3D Dice* ↑				${\mathrm{HD}}_{95}↓$			
	RV	Myo	LV	Average	RV	Myo	LV	Average
Baseline *	$0.525_{\;\pm \;0.184}$	$0.512_{\;\pm \;0.103}$	$0.704_{\;\pm \;0.159}$	$0.581_{\;\pm \;0.107}$	$192.5_{\;\pm \;37.7}$	$188.7_{\;\pm \;24.9}$	$182.8_{\;\pm \;42.4}$	$188.0_{\;\pm \;28.7}$
1 Point	$0.686_{\;\pm \;0.102}$	$0.687_{\;\pm \;0.084}$	$0.762_{\;\pm \;0.051}$	$0.712_{\;\pm \;0.049}$	$16.2_{\;\pm \;12.6}$	$12.5_{\;\pm \;16.9}$	$14.9_{\;\pm \;22.9}$	$14.5_{\;\pm \;14.4}$
2 Points	$0.706_{\;\pm \;0.146}$	$0.716_{\;\pm \;0.064}$	$0.908_{\;\pm \;0.085}$	$0.777_{\;\pm \;0.067}$	$14.9_{\;\pm \;15.1}$	$10.7_{\;\pm \;14.3}$	$6.7_{\;\pm \;11.8}$	$10.8_{\;\pm \;10.7}$
3 Points	$0.751_{\;\pm \;0.134}$	$0.769_{\;\pm \;0.057}$	$0.893_{\;\pm \;0.087}$	$0.804_{\;\pm \;0.065}$	$7.7_{\;\pm \;6.9}$	$8.9_{\;\pm \;6.4}$	$12.7_{\;\pm \;12.5}$	$9.8_{\;\pm \;9.2}$
Scribble	$0.878_{\;\pm \;0.085}$	$0.861_{\;\pm \;0.047}$	$0.923_{\;\pm \;0.070}$	$0.887_{\;\pm \;0.051}$	$6.4_{\;\pm \;10.9}$	$3.7_{\;\pm \;4.0}$	$4.1_{\;\pm \;6.3}$	$4.7_{\;\pm \;5.2}$
Mask	$0.900_{\;\pm \;0.075}$	$0.893_{\;\pm \;0.028}$	$0.941_{\;\pm \;0.048}$	$0.911_{\;\pm \;0.050}$	$4.4_{\;\pm \;4.9}$	$2.4_{\;\pm \;4.0}$	$4.0_{\;\pm \;9.7}$	$3.6_{\;\pm \;6.2}$

* Implemented by the basic scribble-supervised dual-branch siamese network with $L_{\mathrm{DpCE}}$

## Data Availability

The public MRI datasets are available at https://www.creatis.insa-lyon.fr/Challenge/acdc/databases.html and https://zmiclab.github.io/zxh/0/mscmrseg19/data.html, accessed on 1 January 2024.

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
