# Peer review of "Shape-Aware Adversarial Learning for Scribble-Supervised Medical Image Segmentation with a MaskMix Siamese Network: A Case Study of Cardiac MRI Segmentation"

_bioengineering, 2024, doi:10.3390/bioengineering11111146_

Round 1

Reviewer 1 Report

Comments and Suggestions for Authors

In this paper, the authors introduce MaskMixAdv, a shape-aware, scribble-supervised learning framework for medical image segmentation. The framework uses (1) pseudo-label generation and (2) pseudo-label optimization. A dual-branch Siamese network is employed for pseudo-label generation, while a CNN-based discriminator refines these labels by comparing them against external, unpaired masks. The introduced idea and demonstrated results seem interesting.  The following comments are necessary to enhance and clarify the quality of the paper.

1. The abstract should be concise and provide a clearer statement of its impact on the field. It should highlight the significance of the research and its potential implications for medical image segmentation.

2. Why MaskMixAdv is particularly well-suited for weak supervision compared to other augmentation methods. Please clarify it.

3. Please include a paragraph that describe the organization of the paper at the end of the section introduction.

4. Include citations of studies that specifically address scribble-supervised segmentation, particularly in the context of medical imaging.

5. Cite the work that use weakly-supervised methods in general, and how they perform under different settings.

6. Referencing segmentation models that are anatomically-aware, and discussing their limitations in relation to weak supervision.

7. Show the high-level architecture of the MaskMixAdv framework.

8. Explain why the proposed framework is tested on the cardiac MRI dataset instead of other available datasets.

9. Split Figure 1 into separate figures to improve clarity.

10. Figure 2 (The same comment as for Figure 1)

Reviewer 2 Report

Comments and Suggestions for Authors This is an interesting manuscript dealing with the ML techniques and image-analysis methods in applications to i.a. cardiac MRI segmentation. Some of the results are sound and the material itself fits the scope of the current journal very well. I thus support its publication, subject to moderate revisions. The authors should thus address the points listed below.   The entire text is rather long and needs more structure. The problems start already in the abstract, where the authors should be more concise and simply list their main results and achievements, without mentioning any general-knowledge or state-of-the-art subjects/themes.   The introduction is long and interesting, but again it requires much more structure. The authors should clearly describe the general problem here, and then narrow their description and to clearly present the particular/smaller problem they address in the main text.   Towards the end of the introduction the authors should concisely but clearly state what they do, how they do it (methods, approaches, models, etc.), and what exactly do they get as the results. A sectioned plan of the entire paper is to be presented here.   The authors can also discuss somewhat similar ML-based algorithms of classification of elementary features of images performed in other disciplines in recent years. For instance, recently based on the dynamics of local steps of animals on a heterogeneous landscape a ML-based selection algorithm of models of diffusion was examined and proposed, see ref. [DOI: https://doi.org/10.1103/PhysRevResearch.5.043129]. A relation of the Hausdorff distance the authors use in the current submission and the minimal "distance" from the most statistically possible model of diffusion described in the PRR paper above should be discussed. The point here is to present a comparison to other methods of image analysis via employing the ML algorithms and respective optimization procedures.
  In sec. 4.4.2 the authors do describe the efficiency of the proposed algorithm, but it would be also very beneficial to compare the current proposed methods with the already exiting approaches in the literature. Why do we actually need one more method to do the same type of analysis? This should be clarified and also motivated by application examples better!   The authors do provide the definition of the Hausdorff distance in eq. 7, but they do not give any physical explanation what exactly this parameter measures and why namely this measure is the best one to quantify the observed behavior. The authors should improve on this issue in the revised version.

Round 2

Reviewer 2 Report

Comments and Suggestions for Authors

Solid revision, both scientifically and presentation-wise. The material can be accepted.

Author Response

Dear Reviewer,

Thank you very much for your positive evaluation of our revised manuscript.

We are pleased to hear that the revisions have been deemed solid both scientifically and in terms of presentation, and that the material is considered acceptable for publication.

We would like to express our gratitude for the opportunity to contribute to the field and for the guidance provided throughout the review process.

We believe that the improvements made in response to the reviewers' comments have significantly enhanced the quality and clarity of our work.